# Learning To Avoid Negative Transfer in Few-Shot Transfer Learning

## Abstract

Many tasks in natural language understanding require learning relationships between two sequences for various tasks such as natural language inference, paraphrasing and entailment. These aforementioned tasks are similar in nature, yet they are often modeled individually. Knowledge transfer can be effective for closely related tasks, which is usually carried out using parameter transfer in neural networks. However, transferring all parameters, some of which irrelevant for a target task, can lead to sub-optimal results and can have a negative effect on performance, referred to as *negative* transfer.

Hence, this paper focuses on the transferability of both instances and parameters across natural language understanding tasks by proposing an ensemble-based transfer learning method in the context of few-shot learning.

Our main contribution is a method for mitigating negative transfer across tasks when using neural networks, which involves dynamically bagging small recurrent neural networks trained on different subsets of the source task/s. We present a straightforward yet novel approach for incorporating these networks to a target task for few-shot learning by using a decaying parameter chosen according to the slope changes of a smoothed spline error curve at sub-intervals during training.

Our proposed method show improvements over hard and soft parameter sharing transfer methods in the few-shot learning case and shows competitive performance against models that are trained given full supervision on the target task, from only few examples.

## 1 Introduction

Learning relationships between sentences is a fundamental task in natural language understanding (NLU). Given that there is gradience between words alone, the task of scoring or categorizing sentence pairs is made even more challenging, particularly when either sentence is less grounded and more conceptually abstract e.g sentence-level semantic textual similarity and textual inference.

The area of pairwise-based sentence classification/regression has been active since research on distributional compositional semantics that use distributed word representations (word or sub-word vectors) coupled with neural networks for supervised learning e.g pairwise neural networks for textual entailment, paraphrasing and relatedness scoring Mueller & Thyagarajan (2016).

Many of these tasks are closely related and can benefit from transferred knowledge. However, for tasks that are less similar in nature, the likelihood of negative transfer is increased and therefore hinders the predictive capability of a model on the target task. However, challenges associated with transfer learning, such as negative transfer, are relatively less explored explored with few exceptions Rosenstein et al. (2005); Eaton et al. (2008)and even fewer in the context of natural language tasks Pan et al. (2012). More specifically, there is only few methods for addressing negative transfer in deep neural networks Long et al. (2017).

Therefore, we propose a transfer learning method to address negative transfer and describe a simple way to transfer models learned from subsets of data from a source task (or set of source tasks) to a target task. The relevance of each subset per task is weighted based on the respective models validation performance on the target task. Hence, models within the ensemble trained on subsets of a source task which are irrelevant to the target task are assigned a lower weight in the overall ensemble

prediction on the target task. We gradually transition from using the source task ensemble models for prediction on the target task to making predictions solely using the single model trained on few examples from the target task. The transition is made using a decaying parameter chosen according to the slope changes of a smoothed spline error curve at sub-intervals during training. The idea is that early in training the target task benefits more from knowledge learned from other tasks than later in training and hence the influence of past knowledge is annealed. We refer to our method as *Dropping* Networks as the approach involves using a combination of Dropout and Bagging in neural networks for effective regularization in neural networks, combined with a way to weight the models within the ensembles.

For our experiments we focus on two Natural Language Inference (NLI) tasks and one Question Matching (QM) dataset. NLI deals with inferring whether a hypothesis is true given a premise. Such examples are seen in entailment and contradiction. QM is a relatively new pairwise learning task in NLU for semantic relatedness that aims to identify pairs of questions that have the same intent. We purposefully restrict the analysis to no more than three datasets as the number of combinations of transfer grows combinatorially. Moreover, this allows us to analyze how the method performs when transferring between two closely related tasks (two NLI tasks where negative transfer is less apparent) to less related tasks (between NLI and QM). We show the model averaging properties of our negative transfer method show significant benefits over Bagging neural networks or a single neural network with Dropout, particularly when dropout is high (p=0.5). Additionally, we find that distant tasks that have some knowledge transfer can be overlooked if possible effects of negative transfer are not addressed. The proposed weighting scheme takes this issue into account, improving over alternative approaches as we will discuss.

## 2 RELATED WORK

### 2.1 NEURAL NETWORK TRANSFER LEARNING

In transfer learning we aim to transfer knowledge from a one or more source task $\mathcal{T}_s$ in the form of instances, parameters and/or external resources to improve performance on a target task $\mathcal{T}_t$. This work is concerned about improving results in this manner, but also not to degrade the original performance of $\mathcal{T}_s$, referred to as *Sequential Learning*. In the past few decades, research on transfer learning in neural networks has predominantly been parameter based transfer. Yosinski et al. (2014) have found lower-level representations to be more transferable than upper-layer representations since they are more general and less specific to the task, hence negative transfer is less severe. We will later describe a method for overcoming this using an ensembling-based method, but before we note the most relevant work on transferability in neural networks.

Pratt et al. (1991) introduced the notion of parameter transfer in neural networks, also showing the benefits of transfer in structured tasks, where transfer is applied on an upstream task from its sub-tasks. Further to this

Pratt (1993), a hyperplane utility measure as defined by $\theta_s$ from $\mathcal{T}_t$ which then rescales the weight magnitudes was shown to perform well, showing faster convergence when transferred to $\mathcal{T}_t$.

Raina et al. (2006) focused on constructing a covariance matrix for informative Gaussian priors transferred from related tasks on binary text classification. The purpose was to overcome poor generalization from weakly informative priors due to sparse text data for training. The off-diagonals of $\sum$ represent the parameter dependencies, therefore being able to infer word relationships to outputs even if a word is unseen on the test data since the relationship to observed words is known. More recently, transfer learning (TL) in neural networks has been predominantly studied in Computer Vision (CV). Models such as AlexNet allow features to append to existing networks for further fine tuning on new tasks . They quantify the degree of generalization each layer provides in transfer and also evaluate how multiple CNN weights are used to be of benefit in TL. This also reinforces to the motivation behind using ensembles in this paper.

### 2.1.1 TRANSFERABILITY IN NATURAL LANGUAGE

Mou et al. (2016) describe the transferability of parameters in neural networks for NLP tasks. Questions posed included the transferability between varying degrees of "similar" tasks, the transferability

of different hidden layers, the effectiveness of hard parameter transfer and the use of multi-task learning as opposed to sequential based TL. They focus on transfer using hard parameter transfer, most relevantly, between SNLI Bowman et al. (2015) and SICK Marelli et al. (2014). They too find that lower level features are more general, therefore more useful to transfer to other similar task, whereas the output layer is more task specific. Another important point raised in their paper was that a large learning rate can result in the transferred parameters being changed far from their original transferred state. As we will discuss, the method proposed here will inadvertently address this issue since the learning rates are kept intact within the ensembled models, a parameter adjustment is only made to their respective weight in a vote.

Howard & Ruder (2018) have recently popularized transfer learning by transferring domain agnostic neural language models (AWD-LSTM Merity et al. (2017)). Similarly, lexical word definitions have also been recently used for transfer learning O' Neill & Buitelaar (2018), which too provide a model that is learned independent of a domain. This mean the sample complexity for a specific task greatly reduces and we only require enough labels to do label fitting which requires fine-tuning of layers nearer to the output Shwartz-Ziv & Tishby (2017).

## 2.2 PAIRWISE MODEL ARCHITECTURES

Before discussing the methodology we describe the current SoTA for pairwise learning in NLU.

Shen et al. (2017) use a Word Embedding Correlation (WEC) model to score co-occurrence probabilities for Question-Answer sentence pairs on Yahoo! Answers dataset and Baidu Zhidao Q&A pairs using both a translation model and word embedding correlations. The objective of the paper was to find a correlation scoring function where a word vector is given while modelling word co-occurrence given as $C(q_i, \alpha_j) = (v_{q_i}^T / ||v_{q_i}||) \times (\mathbf{M}v_{a_j} / \mathbf{M}||v_{a_j}||)$, where $M$ is a correlation matrix, $v_q$ a word vector from a question and a word vector $v_a$ from an answer. The scoring function was then expanded to sentences by taking the maximum correlated word in answer in a question divided by the answer length.

Parikh et al. (2016) present a decomposable attention model for soft alignments between all pairs of words, phrases and aggregations of both these local substructures. The model requires far less parameters compared to attention with LSTMs or GRUs. This paper uses attention in an SN by proposing attention across hidden layer representations of sentences, in an attempt to mimic how humans compare sentences. Weights are often tied in networks, according to the symmetric property $(\mathcal{S}_i^1, \mathcal{S}_i^2)$.

Yang et al. (2017) have described a character-based intra attention network for NLI on the SNLI corpus, showing an improvement over the 5-hidden layer Bi-LSTM network introduced by Nangia et al. (2017) used on the MultiNLI corpus. Here, the architecture also looks to solve to use attention to produce interactions to influence the sentence encoding pairs. Originally, this idea was introduced for pairwise learning by using three Attention-based Convolutional Neural Networks Yin et al. (2015) that use attention at different hidden layers and not only on the word level. Although, this approach shows good results, word ordering is partially lost in the sentence encoded interdependent representations in CNNs, particularly when max or average pooling is applied on layers upstream.

## 3 METHODOLOGY

In this section we start by describing a co-attention GRU network that is used as one of the baselines when comparing ensembled GRU networks for the pairwise learning-based tasks. We then describe the proposed transfer learning method.

**Co-Attention GRU**   Encoded representations for paired sentences are obtained from $\left(\vec{h}_{T_1}^{(l)}, \vec{h}_{T_2}^{(l)}\right)$ where $\vec{h}^{(l)}$ represents the last hidden layer representation in a recurrent neural network. Since longer dependencies are difficult to encode, only using the last hidden state as the context vector $c_t$ can lead to words at the beginning of a sentence have diminishing effect on the overall representation. Furthermore, it ignores interdependencies between pairs of sentences which is the case for pairwise learning. Hence, in the single task learning case we consider using a cross-attention network as a baseline which accounts for interdependencies by placing more weight on words that are

more salient to the opposite sentence when forming the hidden representation, using the attention mechanism Bahdanau et al. (2014). The $\texttt{softmax}$ function produces the attention weights $\alpha$ by passing all outputs of the source RNN, $h_S$ to the softmax conditioned on the target word of the opposite sentence $h_t$. A context vector $c_t$ is computed as the sum of the attention weighted outputs by $\bar{h}_s$. This results in a matrix $A \in \mathbb{R}^{|S| \times |T|}$ where $|S|$ and $|T|$ are the respective sentence lengths (the max length of a given batch). The final attention vector $\alpha_t$ is used as a weighted input of the context vector $c_t$ and the hidden state output $h_t$ parameterized by a xavier uniform initialized weight vector $W_c$ to a hyperbolic tangent unit.

### 3.1 LEARNING TO AVOID NEGATIVE TRANSFER

Here we describe the two approaches that are considered for accelerating learning and avoiding negative transfer on $\mathcal{T}_t$ given the voting parameters of a learned model from $\mathcal{T}_s$. We first start by describing a method that learns to guide weights on $\mathcal{T}_t$ by measuring similarity between $\theta_{\hat{s}}$ and $\theta_{\hat{t}}$ during training by using moving averages on the slope of the error curve. This is then followed by a description on the use of smoothing splines to avoid large changes due to volatility in the error curve during training.

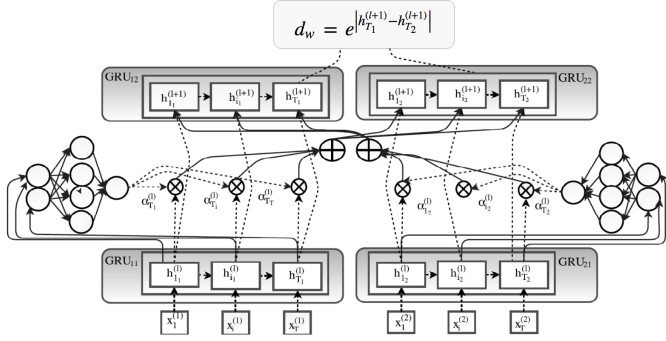

Figure 1: *Cross-Attention GRU-Siamese Network*

***Dropping* Transfer**    Both dropout and bagging are common approaches for regularizing models, the former is commonly used in neural networks. Dropout trains a number of subnetworks by dropping parameters and/or input features during training while also have less parameter updates per epoch. Bagging trains multiple models by sampling instances $\vec{x}_k \in \mathbb{R}^d$ from a distribution $p(\vec{x})$ (e.g uniform distribution) prior to training. Herein, we refer to using both in conjunction as *Dropping*.

The proposed methods is similar to Adaptive Boosting (AdaBoost) in that there is a weight assigned based on performance during training. However, in our proposed method, the weights are assigned based on the performance of each batch after Bagging, instead of each data sample. Furthermore, the use of Dropout promotes sparsity, combining both arithmetic mean and geometric mean model averaging. Avoiding negative transfer with standard AdaBoost is too costly in practice too use on large datasets and is prone to overfitting in the presence of noise Mason et al. (2000). A fundamental concern in TL is that we do not want to transfer irrelevant knowledge which leads to slower convergence and/or suboptimal performance. Therefore, dropping places soft attention based on the performance of each model from $\mathcal{T}_s \rightarrow \mathcal{T}_t$ using a $\texttt{softmax}$ as a weighted vote. Once a target model $f_t$ is learned from only few examples on $\mathcal{T}_t$ (referred to as few-shot learning), the

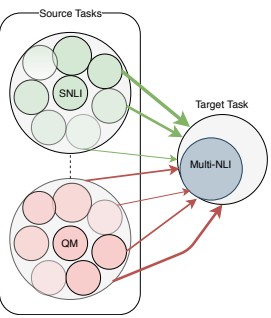

Figure 2: Nodes correspond to models within an ensemble for a given task. Link size= Model weight in target task ensemble prediction.

weighted ensembled models from $\mathcal{T}_s$ can be transferred and merged with the $\mathcal{T}_t$ model. Equation

1 shows the simple weighted vote between models where $N$ is the number of ensembled models each of which have batch size $S$, $\phi$ denotes the softmax function, $z_{s_i}^l = \exp(-|h_{T_1}^l - h_{T_2}^l|)$ and $\bar{a}_s^l$ denotes weighted average output from the ensembles trained on subsets of $\mathcal{T}_s$.

$$\bar{a}_s^l = \sum_{i=1}^{N} \alpha_i \Big( \frac{1}{S} \sum_{s=1}^{S} \phi(z_{s_i}^l) \Big) \quad s.t, \quad \sum_{i=1}^{N} \alpha_i = 1 \tag{1}$$

Equation 2 then shows a straightforward update rule that decays the importance of $\mathcal{T}_s$ *Dropping* networks as the $\mathcal{T}_t$ neural network begins to learn from only few examples. The prediction from few samples $a_t^l$ is the single output from $\mathcal{T}_t^l$ and $\gamma$ is the slope of the error curve that is updated at regular intervals during training. We expect this approach to lead to faster convergence and more general features as the regularization is in the form of a decaying constraint from a related task. The rate of the shift towards the $\mathcal{T}_t$ model is proportional to the gradient of the error $\nabla_{x_{\bar{s}}}$ for a set of mini-batches $x_{\bar{s}}$. In our experiments, we have set the update of the slope to occur every 100 iterations.

$$\hat{y}_t = \gamma \bar{a}_s^l + (1 - \gamma) a_t^l \quad s.t, \quad \gamma = e^{-\delta} \tag{2}$$

The assumption is that in the initial stages of learning, incorporating past knowledge is more important. As the model specializes on the target task we then rely less on incorporating prior knowledge over time. In its simplest form, this can be represented as a moving average over the development set error curve so to choose $\delta_t = \mathbb{E}[\nabla_{[t,t+k]}]$, where $k$ is the size of the sliding window. In some cases an average over time is not suitable when the training error is volatile between slope estimations. Hence, alternative smoothing approaches would include kernel and spline models Eubank (1999) for fitting noisy, or volatile error curves. A kernel $\psi$ can be used to smooth over the error curve, which takes the form of a Gaussian kernel $\psi(\hat{x}, x_i) = \exp\left(-(\hat{x} - x_i)^2 / 2b^2\right)$. Another approach is to use Local Weighted Scatterplot Smoothing (LOWESS) Cleveland (1979); Cleveland & Devlin (1988) which is a non-parametric regression technique that is more robust against outliers in comparison to standard least square regression by adding a penalty term.

Equation 3 shows the regularized least squares function for a set of cubic smoothing splines $\psi$ which are piecewise polynomials that are connected by *knots*, distributed uniformly across the given interval $[0, T]$. Splines are solved using least squares with a regularization term $\lambda \theta_j^2 \; \forall \; j$ and $\psi_j$ a single piecewise polynomial at the subinterval $[t, t + k] \in [0, T]$, as shown in Equation 3. Each subinterval represents the space that $\gamma$ is adapted for over time i.e change the influence of the $\mathcal{T}_s$ *Dropping* Network as $\mathcal{T}_t$ model learns from few examples over time. This type of cubic spline is used for the subsequent result section for *Dropping* Network transfer.

$$\hat{\delta}_{[t]} = \arg\min_{\theta} \sum_{i=1}^{k} \Big( y_i - \sum_{j=1}^{J} \theta_j \psi_j(x_i) \Big)^2 + \lambda \sum_{j=1}^{J} \theta_j^2 \tag{3}$$

The standard cross-entropy (CE) loss is used as the objective as shown in Equation 4.

$$\mathcal{L} = -\frac{1}{N} \sum_{i=1}^{N} \sum_{c=1}^{M} y_{i,c} \log(\hat{y}_{i,c}) \tag{4}$$

This approach is relatively straightforward and on average across all three datasets, 58% more computational time for training 10 smaller ensembles for each single-task was needed, in comparison to a larger global model on a single NVIDIA Quadro M2000 Graphic Processing Unit.

Some benefits of the proposed method can be noted at this point. Firstly, the distance measure to related tasks is directly proportional to the online error of the target task. In contrast, hard parameter sharing does not address such issues, nor does recent approaches that use Gaussian Kernel Density estimates as parameter contraints on the target task O' Neill & Buitelaar (2018). Secondly, although not the focus of this work, the $\mathcal{T}_t$ model can be trained on a new task with more or less classes by adding or discarding connections on the last softmax layer. Lastly, by weighting the models within the ensemble that perform better on $\mathcal{T}_t$ we mitigate *negative transfer* problems. We now discuss some of the main results of the proposed *Dropping* Network transfer.

| Model | MNLI | | | | SNLI | | | | QM | | | |
|---|---|---|---|---|---|---|---|---|---|---|---|---|
| | Train | | Test | | Train | | Test | | Train | | Test | |
| | Acc. / % | LL | Acc. / % | LL | Acc. / % | LL | Acc. / % | LL | Acc. / % | LL | Acc. / % | LL |
| GRU-1 | 91.927 | 0.230 | 68.420 | 1.112 | 89.495 | 0.233 | 77.347 | 0.755 | 84.577 | 0.214 | 78.898 | 0.389 |
| GRU-2 | 90.439 | 0.243 | 68.277 | 1.121 | 89.464 | 0.224 | 79.628 | 0.626 | 86.308 | 0.096 | 77.059 | 0.092 |
| Bi-GRU-2 | 90.181 | 0.253 | 68.716 | 1.065 | 89.703 | 0.226 | 80.594 | 0.636 | 88.011 | 0.108 | 77.522 | 0.267 |
| Co-Attention GRU-2 | 94.341 | 0.183 | 70.692 | 0.872 | 91.338 | 0.211 | 82.513 | 0.583 | 89.690 | 0.088 | 81.550 | 0.218 |
| Ensemble Bi-GRU-2 | 91.767 | 0.260 | 70.748 | 0.829 | 90.091 | 0.218 | 81.650 | 0.492 | 88.481 | 0.177 | 83.820 | 0.194 |

Table 1: Single Task Compositional Similarity Learning Results (shaded values represent best performing models)

# 4 EXPERIMENTAL SETUP

## 4.1 DATASET DESCRIPTION

NLI deals with inferring whether a hypothesis is true given a premise. Such examples are seen in entailment and contradiction. The SNLI dataset Bowman et al. (2015) provides the first large scale corpus with a total of 570K annotated sentence pairs (much larger than previous semantic matching datasets such as the *SICK* Marelli et al. (2014) dataset that consisted of 9927 sentence pairs). As described in the opening statement of McCartney's thesis MacCartney (2009), "the emphasis is on informal reasoning, lexical semantic knowledge, and variability of linguistic expression." The SNLI corpus addresses issues with previous manual and semi-automatically annotated datasets of its kind which suffer in quality, scale and entity co-referencing that leads to ambiguous and ill-defined labeling. They do this by grounding the instances with a given scenario which leaves a precedent for comparing the contradiction, entailment and neutrality between premise and hypothesis sentences.

Since the introduction of this large annotated corpus, further resources for *Multi-Genre NLI* (MultiNLI) have recently been made available as apart of a Shared RepEval task Nangia et al. (2017); Williams et al. (2017). MultiNLI extends a 433k instance dataset to provide a wider coverage containing 10 distinct genres of both written and spoken English, leading to a more detailed analysis of where machine learning models perform well or not, unlike the original SNLI corpus that only relies only on image captions. As authors describe, "temporal reasoning, belief, and modality become irrelevant to task performance" are not addressed by the original SNLI corpus. Another motivation for curating the dataset is particularly relevant to this problem, that is the evaluation of transfer learning across domains, hence the inclusion of these datasets in the analysis. These two NLI datasets allow us to analyze the transferability for two closely related datasets.

*Question Matching* (QM) is a relatively new pairwise learning task in NLU for semantic relatedness, first introduced by the Quora team in the form of a Kaggle competition[1]. The task has implications for Question-Answering (QA) systems and more generally, machine comprehension. A known difficulty in QA is the problem of responding to a question with the most relevant answers. In order to respond appropriately, grouping and relating similar questions can greatly reduce the possible set of correct answers.

## 4.2 TRAINING DETAILS

For single-task learning, the baseline proposed for evaluating the co-attention model and the ensemble-based model consists of a standard GRU network with varying architecture settings for all three datasets. During experiments we tested different combinations of hyperparameter settings. All models are trained for 30,000 epochs, using a dropout rate $p = 0.5$ with Adaptive Momentum (ADAM) gradient based optimization Kingma & Ba (2014) in a 2-hidden layer network with an initial learning rate $\eta = 0.001$ and a batch size $b_T = 128$. As a baseline for TL we use hard parameter transfer with fine tuning on 50% of $X \in \mathcal{T}_s$ of upper layers.

For comparison to other transfer approaches we note previous findings by Yosinski et al. (2014) which show that lower level features are more generalizable. Hence, it is common that lower level features are transferred and fixed for $\mathcal{T}_t$ while the upper layers are fine tuned for the task, as described in Section 2.2. Therefore, the baseline comparison simply transfers all weights from $\theta_s \rightarrow \theta_t$

---

[1]see here: https://www.kaggle.com/c/quora-question-pairs

| | Zero-Shot Hard Parameter Transfer | | | |
| --- | --- | --- | --- | --- |
| | Train | | Test | |
| | Acc. / % | LL | Acc. / % | LL |
| S → M | 60.439 | 0.243 | 61.277 | 1.421 |
| S+Q → M | 62.317 | 0.208 | 62.403 | 1.392 |
| M → S | 74.609 | 0.611 | 71.662 | 0.844 |
| M+Q → S | 74.911 | 0.603 | 68.006 | 0.924 |

Table 2: *Zero-Shot* Hard Parameter Transfer

from a global model instead of ensembles and these parameters as initialization before training on few examples on $\mathcal{T}_t$. Although, negative transfer can occur if the more generalizable lower level representations include redundant or irrelevant examples for the $\mathcal{T}_t$. Instead, here we are allowing the $\mathcal{T}_t$ to guide the lower level feature representations based on a weighted vote in the context of a decaying ensemble-based regularizer.

## 5 RESULTS

The evaluation is carried out on both the rate of convergence and optimal performance. Hence, we particularly analyze the speedup obtained in the early stages of learning. Table 1 shows the results on all three datasets for single-task learning, the purpose of which is to clarify the potential performance if learned from most of the available training data (between 70%-80% of the overall dataset for the three datasets).

The ensemble model slightly outperforms other networks proposed, while the co-attention network produces similar performance with a similar architecture to the ensemble models except for the use of local attention over hidden layers shared across both sentences. The improvements are most notable on MNLI, reaching competitive performance in comparison to state of the art (SoTA) on the RepEval task[2], held by Chen et al. (2017) which similarly uses a Gated Attention Network. These SoTA results are considered as an upper bound to the potential performance when evaluating the *Dropping* based TL strategy for few shot learning.

Figure 3 demonstrates the performance of the zero-shot learning results of the ensemble network which averages the probability estimates from each models prediction on the $\mathcal{T}_t$ test set (few-shot $\mathcal{T}_t$ training set or development set not included). As the ensembles learn on $\mathcal{T}_s$ it is evident that most of the learning has already been carried out by 5,000-10,000 epochs. Producing entailment and contradiction predictions for multi-genre sources is significantly more difficult, demonstrated by lower test accuracy when transferring SNLI → MNLI, in comparison to MNLI → SNLI that performs better relative to recent SoTA on SNLI. Table 2 shows best performance of this hard parameter transfer from $\mathcal{T}_s \rightarrow \mathcal{T}_t$. The QM dataset is not as "similar" in nature and in the zero-shot learning setting the model's weights $a_S$ and $a_Q$ are normalized to 1 (however, this could have been weighted based on a prior belief of how "similar" the tasks are). Hence, it is unsurprising that the QM dataset has reduced the test accuracy given that it is further to $\mathcal{T}_t$ than $S$ is.

The second approach is shown on the LHS of Table 3 which is the baseline few-shot learning performance with fixed parameter transferred from $\mathcal{T}_t$ on the lower layer with fine-tuning of the $2^{nd}$ layer. Here, we ensure that instances from each genre within MNLI are sampled at least 100 times and that the batch of 3% the original size of the corpus is used (14,000 instances). Since SNLI and QM are created from a single source, we did not to impose such a constraint, also using a 3% random sample for testing. Therefore, these results and all subsequent results denoted as *Train Acc. %* refers to the training accuracy on the small batches for each respective dataset. We see improvements that are made from further tuning on the small $\mathcal{T}_t$ batch that are made, particularly on MNLI with a 2.815 percentage point increase in test accuracy. For both SNLI + QM → MNLI and MNLI + QM → SNLI cases final predictions are made by averaging over the class probability estimates before using CE loss.

---

[2]https://repeval2017.github.io/shared/

| | *Few-Shot* Transfer Learning | | | | Dropping-GRU CSES | | | |
|---|---|---|---|---|---|---|---|---|
| | Train | | Test | | Train | | Test | |
| | Acc. / % | LL | Acc. / % | LL | Acc. / % | LL | Acc. / % | LL |
| S → M | 89.655 | 0.248 | 64.897 | 1.696 | 90.439 | 0.243 | 66.207 | 1.721 |
| S+Q → M | 87.014 | 0.376 | 65.218 | 1.255 | 86.649 | 0.317 | 70.703 | 0.576 |
| M → S | 86.445 | 0.260 | 73.141 | 0.729 | 90.181 | 0.253 | 72.716 | 0.615 |
| M+Q → S | 85.922 | 0.281 | 70.541 | 0.911 | 91.783 | 0.228 | 77.926 | 0.598 |

Table 3: *Few-Shot Transfer Learning* with Fixed Lower Hidden GRU-Layer Parameter Transfer From $\mathcal{T}_s$ and Fine-Tuned Upper Layer Trained On $\mathcal{T}_t$ and *Dropping* GRU Between $\mathcal{T}_s \to \mathcal{T}_t$ Using Cubic Spline Error Curve Smoothing

On the RHS, we present the results of the proposed method which transfers parameters from the *Dropping* network trained with the output shown in Equation 2 using a spline smoother with piecewise polynomials (as described in Equation 3). As aforementioned, this approach finds the slope of the online error curve between sub-intervals so to choose $\gamma$ i.e the balance between the source ensemble and target model trained on few examples. In the case with SNLI + QM (ie. SNLI + Question Matching) and MNLI + QM, 20 ensembles are transferred, 10 from each model with a dropout rate $p_d = 0.5$. We note that unlike the previous two baselines methods shown in Table 2 and 3, the performance does not decrease by transferring the QM models to both SNLI and MultiNLI. This is explained by the use of the weighting scheme pro-

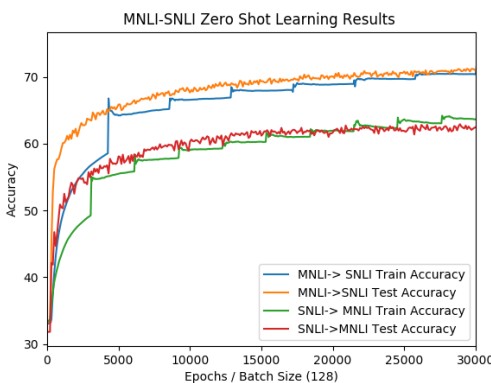

Figure 3: *Zero-Shot Learning Between NLU Tasks*

posed with spline smoothing of the error curve i.e $\gamma$ decreases at a faster rate for $\mathcal{T}_t$ due to the ineffectiveness of the ensembles created on the QM dataset.

In summary, we find transfer of MNLI + QM → SNLI and SNLI+QM → MNLI showing most improvement using the proposed transfer method, in comparison to standard hard and soft parameter transfer. This is reflected in the fact that the proposed method is the only one which improved on SNLI while still transferring the more distant QM dataset. The method for transfer only relies on one additional parameter $\gamma$. We find that in practice using a higher decay rate $\gamma$ (0.9-0.95) is more suitable for closely related tasks. Decreasing $\gamma$ in proportion to the slope of a smooth spline fitted to the online error curve performs better than arbitrary step changes or a fixed rate for $\gamma$ (equivalent to static hard parameter ensemble transfer). Lastly, If a distant tasks has some knowledge transfer they can be overlooked if possible effects of negative transfer are not addressed. The proposed weighting scheme takes this into account, which is reflected on the RHS of Table 3, showing M + Q → S and S + Q → M show most improvement, in comparison to alternative approaches posed in Table 2 where transferring M + Q → S performed worse than M → S.

## 6  CONCLUSION

Our proposed method combines neural network-based bagging with dynamic cubic spline error curve fitting to transition between source models and a single target model trained on only few target samples. We find our proposed method overcomes limitations in transfer learning such as avoiding negative transfer when attempting to transfer from more distant task, which arises during few-shot learning setting. This paper has empirically demonstrated this for learning complex semantic relationships between sentence pairs for pairwise learning tasks. Additionally, we find the co-attention network and the ensemble GRU network to perform comparably for single-task learning.

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
