# OpenReview forum: "Learning To Avoid Negative Transfer in Few Shot Transfer Learning"
_ICLR.cc/2019/Workshop/LLD — Submitted to LLD 2019_

### Official Review · AnonReviewer1 · 2019-04-05
**8 pages, desk reject**

**Rating:** 1
**Confidence:** 3

**Review:**

This paper is longer than the 4 page limit.

---

### Official Review · AnonReviewer2 · 2019-04-07
**Could be interesting but in need of a rewrite**

**Rating:** 2
**Confidence:** 2

**Review:**

The paper investigates a method for transfer learning where the amount of transfer from one task to another is controlled by a dynamic hyperparameter.
Evaluation is performed by combining the SNLI, MultiNLI and Question Matching datasets in various ways using transfer learning.

The work seems to have some interesting ideas, but the paper is lacking in clarity and therefore it is difficult to evaluate the validity and benefit of this approach.

Much of the system description is confusing and difficult to follow. Some examples:
- In Section 2.2, acronym SN used without defining it.
- In Section 3, matrix A and its size is defined, but it is not explained what the values in there represent or how they are then used.
- Section 3.1 says calculations are made based on "voting parameters", which are not mentioned anywhere else in the paper.
- Variables \theta_s and \theta_t are used but not defined.
- GRU-1 and GRU-2 are different configurations in the results tables, but are never defined.
- Tables 2 and 3 use S, M and Q, which are not defined anywhere. I'm guessing these are meant to reference SNLI, MNLI and QM datasets, but explaining this in the caption would be helpful.

Overall, the results and findings are difficult to interpret. For clarity and comparable evaluation, the results tables should contain a non-transfer baseline, a simple fine-tuning baseline, and the SOTA results, in addition to the proposed models. Also, the tables are in need of informative captions about the training conditions.

Section 2.2 describes 3 systems from previous work that are claimed to be SOTA for the tasks that are being investigated. The results section also claims to achieve results that are comparable to SOTA. However, these models are from 2016 and 2017, and there has been quite a bit of work on NLI since then.
The best reported accuracy on SNLI in the paper is 82.5%, whereas current SOTA is at 90+%
https://arxiv.org/pdf/1805.11360v2.pdf
https://arxiv.org/pdf/1901.11504v1.pdf

The best reported accuracy on MultiNLI in the paper is 70.7%, while there are papers reporting 72.2%, 73.9% and 86.7%.
https://arxiv.org/pdf/1804.07461v3.pdf
https://arxiv.org/pdf/1812.01840v2.pdf
https://arxiv.org/pdf/1901.11504v1.pdf

It is not necessary to achieve SOTA in every paper, but comparing to old models and claiming top performance is misleading.

Section 5 says learning was done on "most of the available training data (between 70%-80%). Why not use the whole training data? And why not say exactly how much training data was used?

Section 4.2 says tuning is done on "50% of X \in T_s of upper layers", and it is not clear what is meant by this. That tuning the upper layers was done on 50% of source examples? Why was tuning done on source examples as opposed to target examples? Why only 50%? And why not tune all layers as opposed to only the upper layers?

The concept of "negative transfer" is the main focus of the paper but not really a frequently used term in the field of machine learning, so it should be defined and explained.

The paper is in serious need of proof-reading, as it contains half-finished sentences, many spelling and grammatical errors, and repeated words.

---

### Decision · Program_Chairs · 2019-04-08
**Acceptance Decision**

**Decision:**

Reject

**Comment:**

Desk Reject. The paper severely exceeds the 4 page limit.